# Neutrophil-to-Lymphocyte Ratio Could Predict Outcome in Patients Presenting with Acute Limb Ischemia

**DOI:** 10.3390/jcm10194343

**Published:** 2021-09-24

**Authors:** Maurizio Taurino, Francesco Aloisi, Flavia Del Porto, Martina Nespola, Tommaso Dezi, Chiara Pranteda, Luigi Rizzo, Pasqualino Sirignano

**Affiliations:** 1Vascular Surgery Unit, Department of Molecular and Clinical Medicine, Sant’Andrea Hospital, “Sapienza” University of Rome, Via di Grottarossa, 1035/1039, 00189 Rome, Italy; maurizio.taurino@uniroma1.it (M.T.); martina.nespola@gmail.com (M.N.); dezitommaso@gmail.com (T.D.); chiara.pranteda@gmail.com (C.P.); luigi.rizzo@uniroma1.it (L.R.); 2Internal Medicine Unit, Department of Surgery “Paride Stefanini”, Sant’Andrea Hospital, “Sapienza” University of Rome, Via di Grottarossa, 1035/1039, 00189 Rome, Italy; flavia.delporto@uniroma1.it; 3Vascular Surgery Unit, Department of Surgery “Paride Stefanini”, Sant’Andrea Hospital, “Sapienza” University of Rome, Via di Grottarossa, 1035/1039, 00189 Rome, Italy; pasqualino.sirignano@uniroma1.it

**Keywords:** neutrophil to lymphocyte ratio, acute limb ischemia, amputation, biomarkers, vascular medicine

## Abstract

Introduction: Acute limb ischemia (ALI), classified according to Rutherford’s classification (RC), is a vascular emergency burdened by high rates of mortality and morbidity. The need of new and different prognostic values for ALI has emerged, and, among all, the neutrophil-to-lymphocyte ratio (NLR) has been proven as a strong outcome predictor in vascular disease. The aim of this study is to investigate the role of preoperative NLR in predicting clinical outcomes in patients presenting acute limb ischemia. Material and Methods: A single-center retrospective study was conducted between January 2015 and December 2019. Demographic and clinical characteristics, procedural technical aspects, postoperative and early (up to 30-day) outcomes were recorded. All enrolled patients were categorized into low- and high-NLR at baseline, using a cut-off value of 5. Study outcomes were 30-day all-cause mortality and amputation rates. Results: A total of 177 ALI patients were included in the final analysis (6 RC I, 44 RC IIA, 108 RC IIB, and 19 RC III), 115 males (65%), mean age 78.9 ± 10.4 years. Mean NLR at hospital presentation was 6.65 ± 6.75 (range 0.5–35.4), 108 (61.1%) patients presented a low-NLR, 69 (38.9%) a high-NLR. Immediate technical success was achieved in 90.1% of cases. At 30 days, freedom from amputation and freedom from death rates were 87.1% and 83.6%, respectively. At the univariate analysis, amputation (*p* < 0.0001, OR: 9.65, 95%CI: 3.7–25.19), mortality (*p* = 0.0001, OR: 9.88, 95%CI: 3.19–30.57), and cumulative event rates (*p* < 0.001, OR: 14.45, 95%CI: 6.1–34.21), were significantly different between the two groups according to NLR value. Multivariate analysis showed that a high baseline NLR value was an independent predictor of unfavorable outcomes in all enrolled patients. Consistently, at ROC analysis, a preoperative NLR > 5 was strongly associated with all outcome occurrences. Conclusion: Preoperative NLR value seems to be strongly related to ALI outcomes in this unselected population. The largest series should be evaluated to confirm present results.

## 1. Introduction

Acute limb ischemia (ALI) is defined as a clinical condition characterized by a sudden decrease in arterial perfusion, mostly due to embolism or thrombosis. Regardless of underlying causes, ALI represents a vascular emergency potentially burdened by a high rate of limb- and life-threatening complications [1].

ALI severity and its related prognosis are classically categorized on the base of physical examination and arterial and venous Doppler signals according to the Rutherford’s classification (RC). The four RC grades range from I (viable) to III (irreversible) [2]. The major challenge for vascular specialists is represented by marginally (II A), and immediately threatened (II B) limb for which a prompt diagnosis and an appropriate and effective treatment could greatly determine the outcome [3,4].

Medical and surgical advances have overcome the traditional Fogarty balloon approach and its limitations, allowing a tailored approach for RC II ALI patients [5,6]. Nevertheless, ALI is still associated with a significant mortality and amputation rate up to 40% and 50%, respectively [7].

Consequently, the need of new and different prognostic values has emerged. The neutrophil-to-lymphocyte ratio (NLR; normal range 1–3) is an easy to perform test from the white blood cell count [8,9], reflecting the balance of the neutrophilia of inflammation and the relative lymphopenia of a cortisol-induced stress response. NLR has been tested and proven as a strong mortality predictor in patients with cardiovascular disease or peripheral arterial disease [10,11]. Indeed, the underlying pathophysiology of cardiac and peripheral arterial disease is thought to be the mediation of active inflammatory response [10,12]. The role of neutrophils is represented by numerous biochemical pathways such as releasing of arachidonic acid metabolites and platelet-aggregating factors, cytotoxic oxygen-derived free radicals, and hydrolytic enzymes, such as myeloperoxidase, elastase, various hydrolytic enzymes, and acid phosphatases.

The aim of this study is to investigate the role of preoperative NLR in predicting clinical outcomes in an unselected RC II ALI patients’ population.

## 2. Material and Methods

### 2.1. Study Design

A single-center retrospective study was conducted between January 2015 and December 2019. At tertiary referral hospital admission, all consecutive ALI patients evaluated by the Vascular and Endovascular Surgery Unit of Sant’Andrea Hospital—Sapienza University of Rome (Rome, Italy) were categorized according to the Rutherford Classification into four different grades (RC I, IIA, IIB, III) on the basis of clinical presentation and arterial and venous Doppler signals.

All patients admitted for ALI and submitted to urgent surgical treatment were included in the present study, on an intention-to-treat based-analysis. Exclusion criteria were any concomitant clinical status highly influencing NLR baseline value: chronic kidney disease, liver failure, malignancy, hematologic disease, or inflammatory bowel disease.

Enrolled patients were categorized into low- and high-NLR groups on the basis of their NLR value at baseline. A baseline NLR cut-off value of 5 was selected, according to previous published studies performed in different fields of the vascular diseases [1,10,13].

### 2.2. Data Collection

Demographic and clinical characteristics, procedural technical aspects, perioperative (up to 30-day) outcomes were entered in a dedicated database. Age, sex, arterial hypertension, dyslipidemia, diabetes mellitus, cardiogenic arrhythmias, chronic obstructive pulmonary disease (COPD), anticoagulation therapy, smoking habits, and positive medical history for previous vascular surgery were considered as potentially influencing the outcome and recorded in all patients. All the evaluated parameters were collected and analyzed by only one author (MN).

### 2.3. Preoperative Work-Up and Revascularization Technique

Preoperative work-up consisted of physical examination, blood test (glucose level, haemoglobin, white blood cells, neutrophil count, lymphocyte count, platelet count, serum creatinine, and blood urea nitrogen), and duplex ultrasound scan (DUS) in all patients. Computed tomographic angiography (CTA) was selectively performed to better address patients’ anatomical status (level of arterial occlusion, number of below-the-knee run-off vessels, concomitant aneurysms, grade of atherosclerotic disease). NLR value has been retrospectively evaluated using a peripheral blood sample taken at hospital admission; NLR value was calculated dividing the absolute neutrophil count by the absolute lymphocyte count [12,14].

Revascularization techniques were surgical embolectomy by means a Fogarty balloon catheter, loco-regional transcatheter thrombolysis, and aspiration thrombectomy. The therapeutic approach was chosen basing on ALI aetiology, patients’ general status, and surgeons’ expertise.

### 2.4. Study Outcomes

Primary endpoints were 30-day any-cause mortality rate and 30-day major amputations’ rate (including all above the ankle amputations), and a composite endpoint of both mortality and amputation rates. Outcomes were stratified for RC at hospital admission and NLR value at baseline.

### 2.5. Ethical Approval

The study was conducted according to the guidelines of the Declaration of Helsinki and approved by the Local Ethics Committee of “Sapienza” University of Rome—Policlinico Umberto I Hospital and Sant’Andrea Hospital (Project dentification code384/17). All patients enrolled in the study gave their informed written consent to be submitted for intervention and to be included in the present analysis.

### 2.6. Statistical Analysis

Continuous variables were expressed as mean ± SD and were compared using a paired or unpaired Student’s *t*-test. Categorical variables were expressed as counts and percentages and compared using Fisher exact test or the chi-squared test. Odds ratio and risk ratio were calculated to study the primary endpoint for clinical and procedural variables. A 2-sided value of *p* < 0.05 was considered statistically significant.

A multivariate logistic regression analysis that included variables with *p* < 0.1 was performed to identify independent predictors of amputation, mortality, and a composite endpoint of amputation and mortality. To assess the predictive capacity, receiver operating characteristics (ROC) curves were calculated; cut-off investigation was performed by evaluating the sensitivity and specificity of the ROC curves using the Youden J statistic (J = sensitivity + specificity 1) to d validate the selected threshold. Statistical tests were performed using SPSS 25.0 (IBM Corp, Armonk, NY, USA).

## 3. Results

Two-hundred-one consecutive ALI patients were evaluated during the entire study period. Per protocol, 24 patients were excluded because presenting other concomitant clinical conditions influencing NLR baseline value. Therefore, a total of 177 ALI patients were included in the present analysis: 6 (3.4%) RC I, 44 (24.9%) RC IIA, 108 (61%) RC IIB, and 19 (10.7%) RC III (Figure 1).

One-hundred-fifteen patients were male (65%), mean age was 78.9 ± 10.4 years (42–107); mean NLR at hospital presentation was 6.65 ± 6.75 (range 0.5–35.4). Demographic and clinical characteristics at baseline, as well as performed procedures are reported in Table 1.

Regarding ALI aetiology, 90 patients (50.8%) were found to be affected by in situ atherothrombosis, 73 (41.3%) by cardiac embolism, and 14 (7.9%) by surgical graft thrombosis. Revascularization was performed in 172 (97.2%) patients, while a primary amputation was required in 5 cases. When performed, revascularization was done via surgical embolectomy in 95 patients (53.7%), loco-regional transcatheter thrombolysis in 62 (35%), mechanical thrombectomy in 6 (3.4%), and surgical bypass graft in 9 (5.1%).

Demographic, clinical, and procedural characteristics, except for ALI severity according to RC at hospital admission, were not different between patients with high (>5.0) or low (<5.0) NLR value at baseline (Table 1).

Immediate technical success was achieved in 155 out of 172 patients (90.1%); in 2 cases, a subsequent bypass graft was performed, and in 15 cases, a major amputation was needed (11 above the knee amputations).

At 30-day follow-up, freedom from amputation and freedom from death rates were 87.1% and 83.6%, respectively.

At univariate analysis, amputation, mortality, and combined amputation-mortality rates in the entire cohort of patients were significantly different in high-NLR patients compared with low-NLR (Table 2).

After stratification for RC at presentation, differences between groups were statistically significant for amputation (*p* = 0.02, OR: 15.5; CI95%: 1.51–158.53) and amputation + mortality rates (*p* = 0.003; OR: 15, CI95%: 2.4–93.01), but not for mortality (*p* = 0.15, OR: 6.2; CI95%: 0.50–75.84) in RC IIA patients. In patients presenting with RC IIB, differences between low- and high-NLR groups were statistically significant for all analyzed outcome values: *p* = 0.04, 0.002, and 0.0003 for amputation, mortality, and composite endpoint, respectively. In RCIII patients, no differences were found in mortality (*p* = 0.35) and amputation (*p* = 0.38) rates, while a significant difference was identified between the two groups considering the combined amputation and mortality rate (*p* = 0.001, OR: 19.8; CI95%: 0.61–633.81—Table 2).

Multivariate analysis showed that a high baseline NLR value was an independent predictor of unfavorable outcomes in all enrolled patients, while RC at hospital admission represented an independent predictor of amputations and amputation plus mortality (Table 3).

Consistently, at ROC analysis, a preoperative NLR > 5 was strongly associated with all outcome occurrences, except for mortality in RC IIA patients (Figure 2).

## 4. Discussion

The main result of this study is the potential role of preoperative NLR value, using a cut-off of 5 as an effective prognostic biomarker of clinical outcomes in our unselected series of RC II ALI patients.

In accordance with previous published papers on oncological and vascular diseases, our experience has confirmed the role of the baseline NLR value as an excellent marker for inflammatory processes and a promising risk stratification tool [15,16,17,18,19,20]. Previous published studies evaluated the prognostic role of preoperative NLR values in vascular disease, and the importance of establishing a unique cut-off [20,21]. Kordzazdeh et al. reported that a preoperative NLR > 5 could be considered as an independent predictor of 30-day mortality in patients treated for abdominal aortic aneurysms (AAA), irrespective of age, gender, comorbidities, AAA size, blood loss, and length of hospital stay [22]. Similarly, Tokgoz et al. described an NLR > 5.67 as a strong predictor of mortality after acute ischemic stroke (sensitivity 81.7%, and specificity 65.8%) [23]. Moreover, in patients with chronic critical limb ischemia, preoperative NLR > 5 was found to be significantly related to higher mortality rate during 5-year follow up [8,13]. Lastly, Tasoglu et al. reported a strong correlation between preoperative NLR and amputation rate in ALI patients treated by Fogarty embolectomy. In their experience a NLR > 5.2 has been shown to be an independent risk factor for amputation during follow-up [20]. More recently, Pasqui et al. suggested that a cut-off of 5.57 could predict mortality, and a value higher than 6.66 could be associated with amputation rate after ALI surgical or endovascular treatment [24].

Given the absence of a well-defined NLR cut-off value, in the present study, a unique cut-off value of 5 was selected, validated by ROC curve analysis, and successfully adopted to categorize ALI patients (Table 2, Figure 1). At univariate analysis, a NLR greater than 5 was strongly associated with adverse clinical outcomes at 30-day follow-up. Indeed, 30-day amputation rate (5.5% vs. 36%, *p* < 0.001), 30-day mortality rate (3.7 vs. 27.5%, *p* = 0.001), and combined amputation-mortality rate (7.4% vs. 53.6% *p* < 0.0001) were significantly higher in patients presenting a baseline NLR value > 5. Consistently, multivariate analysis showed that NLR value was an independent predictor of adverse event occurrence during follow-up, while RC II was found to be independently associated with amputation occurrence.

Furthermore, prognostic relevance of preoperative NLR value has also been confirmed by ROC analysis for 30-day amputation rate (area under the curve of 0.82, specificity of 80%, sensitivity of 70%), 30-day mortality rate (area under the curve of 0.77, specificity of 82%, sensitivity of 68.5%), and for a 30-day combined mortality-amputation rate (area under the curve of 0.84, specificity of 82%, sensitivity of 75%).

All those findings were confirmed, even under-categorizing ALI patients in Rutherford Class IIA and IIB, except for mortality rate in IIA group. It is noteworthy that patients presenting in RC I or III showed a different trend: RC I cases were more prone to have a favorable outcome; conversely, RC III more frequently experienced an adverse event occurrence, regardless baseline NLR value (Table 2).

Undoubtedly, atherosclerotic disease and chronic inflammation are strictly intertwined. High white blood cell values have been associated with negative outcomes in patients with arterial diseases: neutrophils have a strong effect on atherosclerotic plaque evolution, while lymphopenia is the most common inflammatory marker that appeared in response to an increased corticosteroid path activity secondary to stress and inflammation. Consequently, the use of ratios represents a more informative tool with respect to single cell count, neutrophilia, and lymphocytopenia. From a speculative point of view, it should be underlined, as the inflammatory imbalance expressed by NLR could explain the remarkable difference in outcomes despite similar RC presentation [10,12]. Moreover, acute limb ischemia represents a medical and surgical challenge, and decision-making remains difficult even after the developing and diffusion of endovascular therapy. Our data, according to previously published experience, confirm the necessity of a tailored approach, possibly reserving a more aggressive strategy for those presenting an elevated NLR baseline value [24].

Although the real predictive value of our results is supported by significant statistical significance, the present study has several limitations. The first one is the design of the study: a retrospective single-arm registry, conducted on a relatively small cohort, which is not randomized and does not allow to compare the results with a control patient population. Moreover, the retrospective nature of this study did not allow the performance of subgroup analysis on respect of intra- and post-operative pharmacological therapy.

Otherwise, these data could be considered hypothesis-generating to design a large-scale clinical trial to definitively investigate NLR role in acute limb ischemia. Potentially, NLR value could benefit the preoperative identification of patients via a more aggressive approach combining revascularization to intraoperative thrombolysis and other medical adjuncts, such as oral anticoagulation, anti-inflammatory therapy, and prostaglandin infusion.

In conclusion, our experience, although limited, seems to suggest a role for baseline NLR value in predicting adverse event occurrence after treatment in ALI patients. Undeniably, our data should be validated by a randomized trial, prospectively evaluating results with a proper control patient population.

## Figures and Tables

**Figure 1 jcm-10-04343-f001:**
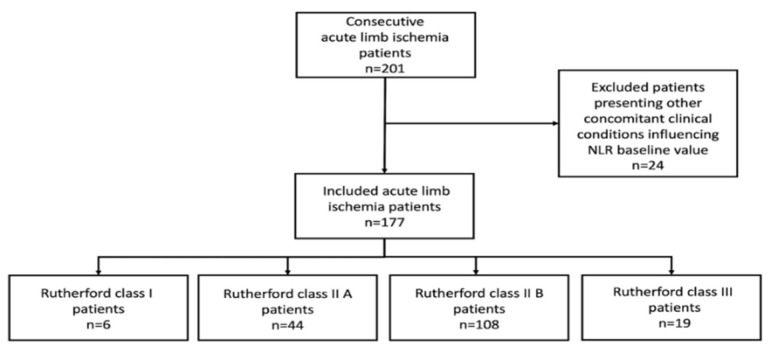
Enrollment flowchart.

**Figure 2 jcm-10-04343-f002:**
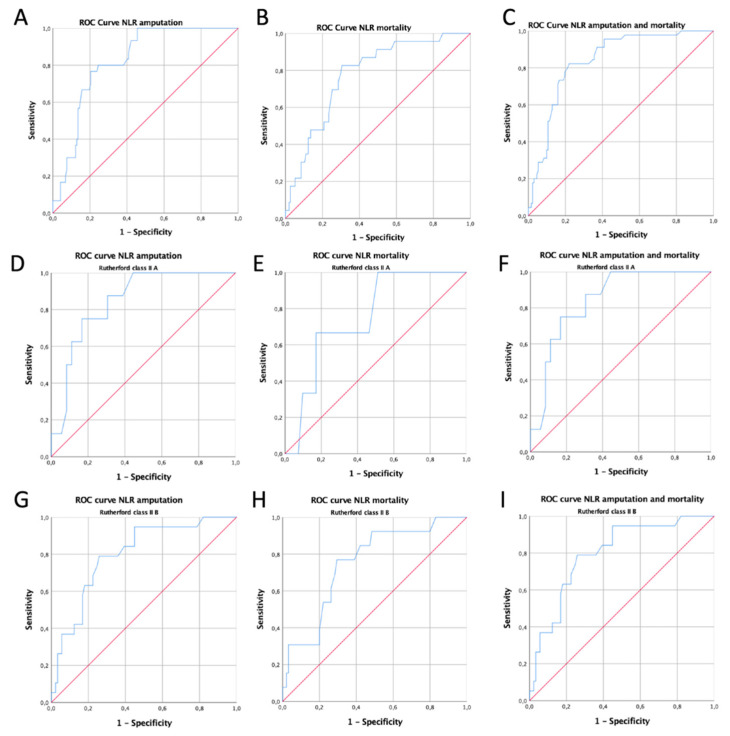
ROC curve analysis (**A**) for NLR with respect to amputation rate in all patients (c-statistic 0.823), (**B**) for NLR with respect to mortality in all patients (c-statistic 0.776), (**C**) for NLR with respect to amputation + mortality rate in all patients (c-statistic 0.840), (**D**) for NLR with respect to amputation rate in RC IIA patients (c-statistic 0.845), (**E**) for NLR with respect to mortality in RC IIA patients (c-statistic 0.752), (**F**) for NLR with respect to amputation + mortality rate in RC IIA patients (c-statistic 0.845), (**G**) for NLR with respect to amputation rate in RC IIB patients (c-statistic 0.798), (**H**), for NLR with respect to mortality in RC IIB patients (c-statistic 0.751), and (**I**) for NLR with respect to amputation + mortality rate in RC IIB patients (c-statistic 0.798).; ROC: receiver-operating characteristic; NLR: neutrophil-to-lymphocyte ratio; RC: Rutherford class.

**Table 1 jcm-10-04343-t001:** Demographic, clinical, and procedural characteristics of all patients included in the analysis and of the two sub-groups evaluated according to low- and high-LNR baseline value.

	All Patients	Low-NLR	High-NLR	*p* (OR; CI95%)
RC I	6/177 (3.4%)	6/108 (5.6%)	0/69 (0%)	<0.001
RC IIA	44/177 (24.9%)	32/108 (29.6%)	12/69 (17.4%)	
RC IIB	108/177 (61%)	67/108 (62%)	41/69 (59.4%)	
RC III	19/177 (10.7%)	3/108 (2.8%)	16/69 (23.2%)	
Male sex	115/177 (65%)	67/108 (62%)	48/69 (69.6%)	0.305 (0.71; 0.37–1.36)
Age (mean ± SD)	78.9 ± 10.4	77.8 ± 11.1	80.6 ± 9.3	0.065
Atrial fibrillation	45/177 (25.4%)	24/108 (22.2%)	21/69 (30.4%)	0.22 (0.65; 0.32–1.29)
Arterial hypertension	145/177 (81.9%)	87/108 (80.6%)	58/69 (84.1%)	0.55 (0.78; 0.35–1.75)
Dyslipidaemia	54/177 (30.5%)	28/108 (25.9%)	26/69 (37.7%)	0.09 (0.57; 0.3–1.1)
Diabetes mellitus	70/177 (39.5%)	38/108 (35.2%)	32/69 (46.4%)	0.13 (0.62; 0.33–1.16)
ALI aetiology				
Arterial thrombosis	90/177 (50.8%)	56/108 (51.9%)	34/69 (49.3%)	0.923
Cardiac embolism	73/177 (41.3%)	44/108 (40.7%)	29/69 (42%)	
Graft thrombosis	14/177 (7.9%)	8/108 (7.4%)	6/69 (8.7%)	
Performed primary procedure				
Fogarty embolectomy	95/177 (53.7%)	63/108 (58.4%)	32/69 (46.5%)	0.053
Fibrinolysis	62/177 (35%)	36/108 (33.3%)	26/69 (37.6%)	
By-pass	9/177 (5.1%)	5/108 (4.6%)	4/69 (5.8%)	
Mechanical thrombectomy	6/177 (3.4%)	4/108 (3.7%)	2/69 (2.9%)	
Major amputation	5/177 (2.8%)	0/108 (0%)	5/69 (7.2%)	

ALI: acute limb ischemia; NLR: neutrophil-to-lymphocyte ratio; RC: Rutherford class.

**Table 2 jcm-10-04343-t002:** Univariate analysis on RC and NLR at presentation and new adverse event occurrence during the study period.

	Amputation	Mortality	Amputation and Mortality
low-NLR vs. high-NLR in all enrolled patients	6/108 (5.5%) vs. 25/69 (36.23%) *p* < 0.0001 OR:9.65; CI95%:3.7–25.19	4/108 (3.7%) vs. 19/69 (27.5%) *p* = 0.0001 OR:9.88; CI95%:3.19–30.57	8/108 (7.4%) vs. 37/69 (53.6%) *p* < 0.001 OR:14.45; CI95%:6.10–34.21
low-NLR vs. high-NLR in RC IIA patients	1/32 (3.1%) vs. 4/12 (33.2%) *p* = 0.02 OR:15.5; CI95%:1.51–158.53	1/32 (3.1%) vs. 2/12 (16.6%) *p* = 0.15 OR:6.2; CI95%:0.50–75.84	2/32 (6.2%) vs. 6/12 (50%) *p* = 0.003 OR:15; CI95%:2.4–93.01
low-NLR vs. high-NLR in RC IIB patients	3/67 (4.5%) vs. 7/41 (17.1%) *p* = 0.04 OR:4.39; CI95%:1.06–18.08	3/67 (4.5%) vs. 11/41 (26.8%) *p* = 0.002 OR:7.8; CI95%:2.03–30.1	4/67 (6%) vs. 15/41 (36.6%) *p* = 0.0003 OR:9.08; CI95%:2.75–29.98
low-NLR vs. high-NLR in RC III patients	2/3 (66.6%) vs. 14/16 (87.5%) *p* = 0.38 OR:3.5; (CI95%:0.20–58.7)	0/3 (0%) vs. 6/16 (37.5%) *p* = 0.35 OR:4.33; CI95%:0.19–98.18	2/3 (66.6%) vs. 16/16 (100%) *p* = 0.001 OR:19.8; CI95%:0.61–633.81

NLR: neutrophil-to-lymphocyte ratio; RC: Rutherford class.

**Table 3 jcm-10-04343-t003:** Multivariate analysis on new adverse event occurrence during the entire study period.

	Amputation		Mortality		Amputation and Mortality	
	*p*	(OR; CI95%)	*p*	(OR; CI95%)	*p*	(OR; CI95%)
RC IIA	0.0020	1.91; 0.02–0.42	0.2923	0.40; 0.07–2.18	0.0035	1.30; 0.02–0.48
RC IIB	<0.0001	2.50; 0.01–0.22	0.6662	1.74; 0.19–2.86	0.0002	2.08; 0.02–0.30
Male sex	0.0748	3.09; 0.89–10.71	0.6592	0.79; 0.28–2.22	0.1152	2.27; 0.81–6.29
Age > 80	0.7787	1.16; 0.40–3.35	0.9976	0.99; 0.36–2.74	0.6064	0.78; 0.31–1.95
High NLR	0.0002	9.79; 2.99–31.97	0.0006	7.71; 2.40–24.74	<0.0001	15.09; 5.44–41.84
Atrial fibrillation	0.9402	0.95; 0.26–3.36	0.7911	1.16; 0.37–3.56	0.7693	1.16; 0.41–3.32
Arterial hypertension	0.9179	1.08; 0.23–4.94	0.9694	1.02; 0.23–4.46	0.6756	1.31; 0.36–4.83
Dyslipidaemia	0.6277	1.33; 0.41–4.25	0.9732	0.98; 0.32–2.95	0.8214	0.88; 0.31–2.50
Diabetes mellitus	0.9060	1.06; 0.36–3.11	0.6576	1.27; 0.43–3.71	0.4164	1.46; 0.58–3.67
Arterial thrombosis	0.8647	1.17; 0.17–7.80	0.9493	0.94; 0.15–5.71	0.5888	0.63; 0.11–3.35
Cardiac embolism	0.6238	0.61; 0.08–4.33	0.9304	1.08; 0.17–6.71	0.3864	0.46; 0.08–2.61

NLR: neutrophil-to-lymphocyte ratio; RC: Rutherford class.

## Data Availability

The data presented in this study are available on request from the corresponding author. The data are not publicly available due to privacy and ethical reasons.

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
