# Peer review of "Neutrophil-to-Lymphocyte Ratio Could Predict Outcome in Patients Presenting with Acute Limb Ischemia"

_jcm, 2021, doi:10.3390/jcm10194343_

Round 1

Reviewer 1 Report

Despite keeping a little excessive use of acronyms in the abstract, I consider myself able to publish

Author Response

Thank you for your kind revision and comment

Reviewer 2 Report

Line 56 has a spelling error 

This is a novel and interesting manuscript which has been improved from the previous submission which I reviewed.  As you clearly outline, there are obvious limitations to a single center retrospective study but nonetheless this provides the ground work for future investigation.

Author Response

Thank you for your kind revision and comment, the line 56 spelling error has been fixed 

This manuscript is a resubmission of an earlier submission. The following is a list of the peer review reports and author responses from that submission.

Round 1

Reviewer 1 Report

Excellent manuscript on an important topic.  The only limitations are the single center study design and small cohort which are clearly outlined and acknowledged by the authors

Author Response

Excellent manuscript on an important topic.  The only limitations are the single center study design and small cohort which are clearly outlined and acknowledged by the authors.

Thank you for your kind words.

Reviewer 2 Report

“NEUTROPHIL TO LYMPHOCYTE RATIO COULD PREDICT OUTCOME IN PATIENTS PRESENTING WITH ACUTE LIMB ISCHEMIA” it is a work that focuses on a current and pertinent theme with relevance for publication, whose title is informative and relevant and the objective clear. The methodology is sufficiently described using scientific language. The presentation of the results is clear, however the introduction to the discussion could be improved in order to enrich the discussion. The following changes are recommended:

Abstract

There is an excessive use of acronyms that makes it difficult to read and understand the text, I suggest that in the summary the use of acronyms should be summarized in the presentation of results, namely in statistics.

Keyword

I do not understand and do not advise the duplicate use of "Neutrophil to Lymphocyte Ratio" with "NLR" and "acute limb ischemia" with "ALI", as they mean the same, they are just acronyms. For this reason, it would be advisable to use key words contained in Mesh.

Introduction

I suggest a short paragraph on the pathophysiological significance of neutrophils and lymphocytes in acute ischemic disease.

Materials and Methods

Since there are more degrees of classification, wouldn't it have been more interesting to include all individuals, that is, with CR I, IIA, IIB and III, thus allowing to be able to compare the variation of the NLR in the different degrees of the disease?

Line 60 and 61 - “A single-center study was retrospectively conducted using a prospectively compiled computerized database between January 2015 and December 2019.“ I suggest that the type of study be clarified: retrospective or prospective.

It should be mentioned if the evaluation of the various parameters evaluated and variables collected was carried out by one or more people, as this will influence the error.

The variables registered in the diagnostic tests were not described, except for blood tests

The groups created in the study, used in the statistical analysis and referred to in the results have not been properly defined in this section, are described almost at the end of the results, and there is already a previous reference to them

The confidence intervals of the tests are not mentioned.

Results

How was the cutoff point selected?

Incorrect use of capital letters within the tables referred to in the results.

Line 114 - Correct the phrase “Ninety-seven patients were male males (63.8%),“

The presentation of Table 3 must be revised, as the way the results are presented is very confusing.

Discussion

Line 173 to 176 - This topic, which is already known on the topic, must be included in the introduction in order to be discussed

Table and Figure Legende

Are there after the discussion for what reason?

References

They must all appear in the same type and font size and line spacing as the rest of the text.

Author Response

“NEUTROPHIL TO LYMPHOCYTE RATIO COULD PREDICT OUTCOME IN PATIENTS PRESENTING WITH ACUTE LIMB ISCHEMIA” it is a work that focuses on a current and pertinent theme with relevance for publication, whose title is informative and relevant and the objective clear. The methodology is sufficiently described using scientific language. The presentation of the results is clear, however the introduction to the discussion could be improved in order to enrich the discussion.

Thank you for your comments.

The following changes are recommended:

Abstract

There is an excessive use of acronyms that makes it difficult to read and understand the text, I suggest that in the summary the use of acronyms should be summarized in the presentation of results, namely in statistics.

Abstract was extensively rewritten

Keyword

I do not understand and do not advise the duplicate use of "Neutrophil to Lymphocyte Ratio" with "NLR" and "acute limb ischemia" with "ALI", as they mean the same, they are just acronyms. For this reason, it would be advisable to use key words contained in Mesh.

Keywords were rewritten according to your suggestions.

Introduction

I suggest a short paragraph on the pathophysiological significance of neutrophils and lymphocytes in acute ischemic disease.

Thank you for your suggestion: a new paragraph was added in the revised text

Materials and Methods

Since there are more degrees of classification, wouldn't it have been more interesting to include all individuals, that is, with CR I, IIA, IIB and III, thus allowing to be able to compare the variation of the NLR in the different degrees of the disease?

Although we agree with your statement, as this is an initial study we preferred to focus only on the portion of patients with the most uncertain prognosis. At a later stage, it will be interesting to re-evaluate the data also by considering patients in Rutherford Class I and III

Line 60 and 61 - “A single-center study was retrospectively conducted using a prospectively compiled computerized database between January 2015 and December 2019.“ I suggest that the type of study be clarified: retrospective or prospective.

The entire sentence was rewritten according to your suggestion.

It should be mentioned if the evaluation of the various parameters evaluated and variables collected was carried out by one or more people, as this will influence the error.

The same physician evaluated and collected all the parameters, this concept is now clearly expressed in the revised text.  

The variables registered in the diagnostic tests were not described, except for blood tests.

Material and methods section was extensively rewritten, also including more details on performed perioperative diagnostic tests.

The groups created in the study, used in the statistical analysis and referred to in the results have not been properly defined in this section, are described almost at the end of the results, and there is already a previous reference to them. Material and Method section has been rewritten to better explain this concept.

The confidence intervals of the tests are not mentioned.

Material and methods section was extensively rewritten, according with your suggestions.

Results

How was the cutoff point selected?

Cut-off point of value was selected accordingly with previous published studies in different filed of cardiovascular vascular disease and validated using ROC curve analysis. Material and method section was rewritten in order to better clarify this concept. Thank you for your question.

Incorrect use of capital letters within the tables referred to in the results.

Results section was modified in accordance with your comment.

Line 114 - Correct the phrase “Ninety-seven patients were male males (63.8%),“ The phrase was corrected in the revised text

The presentation of Table 3 must be revised, as the way the results are presented is very confusing.

All table were redrawn in the revised text.

Discussion

Line 173 to 176 - This topic, which is already known on the topic, must be included in the introduction in order to be discussed.

Thank you for the suggestion; introduction and discussion sections were both accordingly modified in the revised text.

Table and Figure Legend

Are there after the discussion for what reason?

Figures and Tables were embedded in the text, according to your suggestion and Instruction for Authors.

References

They must all appear in the same type and font size and line spacing as the rest of the text.

References’ list was rearranged, according to your suggestion

Reviewer 3 Report

The authors examined the prognostic value of Neutrophil to Lymphocyte (NTL) in acute lower limb ischemia (ALI) for all-cause mortality and lower limb amputation. The optimal cut-offs for these clinical events were determined, and the two groups were divided into two groups. The odds ratios for the incidence of events were calculated, and high NTL was found to be associated with adverse clinical outcomes.

The most critical issue in this study is a novelty. As the author mentioned, previous studies reported that NLR after embolectomy was a prognostic factor.

Furthermore, I regret to say that there are many careless mistakes in the description of the paper, and it does not meet the quality of the publication.

Major

  • How many patients were recruited, excluded, and finally analyzed was not shown; please show the Enrollment Flowchart in Figure
  • For better understanding, add a subsection to the method to organize it. (e.g., Study Participants, Data Collection, Revascularization Technique, and so on)
  • Were technical failure cases included in the analysis? Technical failure was not listed in the exclusion criteria.
  • The description of the primary endpoint was missing. In addition, it did not compare the incidence of High-NLR and Low-NLR.
  • In Table 3, the incidence and odds ratios of the primary endpoint were listed together, which is difficult to see. Please consider that the incidence of the primary endpoint should be described separately from the odds ratio in a table or bar graph.
  • Focusing on the odds ratio (Table 3), there seemed to be an interaction between High-NLRL and RC IIb for mortality. Have you calculated the interaction between them? For better discussion, it would be better to show the background comparison between RCIIA and RCIIB in the supplement file.
  • The authors used the Youden index to determine the cut-off, but was it true that the best cut-off value were all the same for all ROC curves?
  • The Institutional Review Board Statement is the template itself; please describe the ethical statement. In this situation, Conflicts of Interest and Founding information are also doubtful.
  • In the 1st line of Discussion, was it correct "The mean finding of this study”? “The main finding of this study” should be correct.
  • The most critical issue in this study is novelty. As the author mentioned in the Discussion, Tasoglu et al. reported that NLR after embolectomy was a prognostic factor. For a more sophisticated report, it would seem worthwhile to compare the NLR before and after embolectomy and examine the change in its prognostic value.
  • The odds ratio was computed, but the method of analysis was not described in Statistical Analysis. Why did they not perform multivariate analysis or adjustment?
    The results of the study would be more convincing if the logistic model for the primary endpoint (the number of composite endpoint of all-cause death and amputation can be sufficient to analysis) were used and adjusted for the summed risk score.
  • The interaction between RC and NLR was discussed, however, a more sophisticated Discussion would be desired based on the results of the abovementioned analysis.

Minor

  • (Table 1) The percentages of categorical variables should be expressed as integers for percentages of 10 or more, and to one decimal place for percentages of less than 10%. Were there any missing values at all? I think it would be better to specify the denominator as well. (e.g. 97/152 (64%), 9/152 (5.9%).)
  • (Table 1; Table 2; Title of tables) Carefully describe the capitalization. The e in embolism in Cardiac embolism is lower case, but the T in Graft Thrombosis is upper case T. In general, the second and subsequent words would not be capitalized. Capitalization was not consistent throughout the paper. Anyway, keep the same descriptions.
  • (Figure 1) The "1-Specificity" in the X-axis title of Figure 1 E is hidden.
  • How about combining Table 1 and Table 2 into one?
  • (Table) NS is not preferred. Please show the P value.
  • Embed Tables and Figures in the relevant sections. (See author instructions).

Author Response

The authors examined the prognostic value of Neutrophil to Lymphocyte (NTL) in acute lower limb ischemia (ALI) for all-cause mortality and lower limb amputation. The optimal cut-offs for these clinical events were determined, and the two groups were divided into two groups. The odds ratios for the incidence of events were calculated, and high NTL was found to be associated with adverse clinical outcomes.

The most critical issue in this study is a novelty. As the author mentioned, previous studies reported that NLR after embolectomy was a prognostic factor.

Furthermore, I regret to say that there are many careless mistakes in the description of the paper, and it does not meet the quality of the publication.

Major

How many patients were recruited, excluded, and finally analyzed was not shown; please show the Enrollment Flowchart in Figure.

According with your suggestion, an enrollment Flowchart in Figure (Figure 1) was added in the revised manuscript.

For better understanding, add a subsection to the method to organize it. (e.g., Study Participants, Data Collection, Revascularization Technique, and so on)

Material and Method section was extensively revised to address your suggestion

Were technical failure cases included in the analysis? Technical failure was not listed in the exclusion criteria.

Technical failure was not an exclusion criterion for the study. Now this concept it clearly reported in the revised text.

The description of the primary endpoint was missing. In addition, it did not compare the incidence of High-NLR and Low-NLR.

Study endpoints were rewritten.

In Table 3, the incidence and odds ratios of the primary endpoint were listed together, which is difficult to see. Please consider that the incidence of the primary endpoint should be described separately from the odds ratio in a table or bar graph.

Table was redrawn according with your suggestions.

Focusing on the odds ratio (Table 3), there seemed to be an interaction between High-NLR and RC IIb for mortality. Have you calculated the interaction between them? For better discussion, it would be better to show the background comparison between RCIIA and RCIIB in the supplement file.

As far as we can agree with your comment, this type of analysis has not been carried out.

The authors used the Youden index to determine the cut-off, but was it true that the best cut-off value were all the same for all ROC curves?

ROC Curves were used to validate the selected cut-off NLR value of 5. Statistical Analysis paragraph in Material and Method was rewritten to better clarify this point according with your comments.

The Institutional Review Board Statement is the template itself; please describe the ethical statement. In this situation, Conflicts of Interest and Founding information are also doubtful.

Revised text was modified clearly reporting Institutional Review Board Statement.

In the 1st line of Discussion, was it correct "The mean finding of this study”? “The main finding of this study” should be correct.

It was a spelling error. Sentence was corrected

The most critical issue in this study is novelty. As the author mentioned in the Discussion, Tasoglu et al. reported that NLR after embolectomy was a prognostic factor.

Thank you for your valuable comment. However, regarding the novelty of our data, we should accept that there is no consensus about the NLR threshold to be used. All those, despite Tasoglu and more recently Pasqui (as added in discussion) reported similar results. In present study, we tried to validate a simple and unique cut-off value to be used for all outcomes (mortality, amputation, and mortality + amputation).

For a more sophisticated report, it would seem worthwhile to compare the NLR before and after embolectomy and examine the change in its prognostic value.

Unfortunately, the retrospective nature of our study limits the possibility to evaluate post-operative data. NLR values were obtained from a single pre-operative blood sample, and subsequent data were not taken because of the impossibility of retrieving follow-up laboratory data.

The odds ratio was computed, but the method of analysis was not described in Statistical Analysis. Why did they not perform multivariate analysis or adjustment?

Statistical analysis paragraph was rewritten according to your suggestions and comments.

The results of the study would be more convincing if the logistic model for the primary endpoint (the number of composite endpoint of all-cause death and amputation can be sufficient to analysis) were used and adjusted for the summed risk score.

Thank you for your valuable suggestions. However, the present is a preliminary study aiming to validate a cut-off NLR value. We aim to perform such kind of analysis in a larger subsequent experience.

The interaction between RC and NLR was discussed, however, a more sophisticated Discussion would be desired based on the results of the abovementioned analysis.

Thank you for suggestion. Discussion was extensively modified.

Minor

(Table 1) The percentages of categorical variables should be expressed as integers for percentages of 10 or more, and to one decimal place for percentages of less than 10%. Were there any missing values at all? I think it would be better to specify the denominator as well. (e.g. 97/152 (64%), 9/152 (5.9%).)

Tables were redrawn according with your suggestions.

(Table 1; Table 2; Title of tables) Carefully describe the capitalization. The e in embolism in Cardiac embolism is lower case, but the T in Graft Thrombosis is upper case T. In general, the second and subsequent words would not be capitalized. Capitalization was not consistent throughout the paper. Anyway, keep the same descriptions.

Capitalization was corrected were needed.

(Figure 1) The "1-Specificity" in the X-axis title of Figure 1 E is hidden.

Figure (now figure 2) was re-edited according to your suggestion.

How about combining Table 1 and Table 2 into one?

Tables 1 and 2 were merged, as requested

(Table) NS is not preferred. Please show the P value.

Tables were redrawn according with your suggestions

Embed Tables and Figures in the relevant sections. (See author instructions).

Figures and Tables were embedded in the text, according to your suggestion and Instruction for Authors.

Reviewer 4 Report

This present manuscript provides retrospective data on the neutrophil to lymphocyte ratio as a promising outcome predictor in patients with acute limb ischemia. 

The issue dealt with in this present study is of great importance for the scientific community and in particular for clinicians treating patients with ALI.

This reviewer would appreciate further explanation on the importance of identifying patients with poor prognosis. What is the influence on diagnostic and therapeutic strategies? How should patients categorized to have a poor prognosis due to an elevated NLR be treated? Why should clinicians focus on the NLR in daily routine? What is the advantage of the NLR as compared to other outcome parameters? Consider to mention other prognostic variables that are currently being discussed in available literature.

Please consider to adapt the expression for statistical results as OR(95%CI).

Extensive language check for style, spelling and grammar is required! Please take care for consistency using "." or "," for decimal numbers. The manuscript appears a bit "sloppy". Some corrections and suggestions are provided in the PDF file as attached below. 

Author Response

This present manuscript provides retrospective data on the neutrophil to lymphocyte ratio as a promising outcome predictor in patients with acute limb ischemia. The issue dealt with in this present study is of great importance for the scientific community and in particular for clinicians treating patients with ALI.

Thank you for your kind words and comments.

This reviewer would appreciate further explanation on the importance of identifying patients with poor prognosis. What is the influence on diagnostic and therapeutic strategies? How should patients categorized to have a poor prognosis due to an elevated NLR be treated? Why should clinicians focus on the NLR in daily routine? What is the advantage of the NLR as compared to other outcome parameters? Consider to mention other prognostic variables that are currently being discussed in available literature.

Thank you for your suggestion. A dedicated paragraph was added in discussion to better underline the clinical implication of our findings.

Please consider to adapt the expression for statistical results as OR(95%CI).

Thank you for your suggestion. Revised tables were modified as you requested.

Extensive language check for style, spelling and grammar is required! Please take care for consistency using "." or "," for decimal numbers. The manuscript appears a bit "sloppy". Some corrections and suggestions are provided in the PDF file as attached below.

Thank you, the entire paper was revised according to your comments and suggestions

Round 2

Reviewer 3 Report

The manuscript has been much improved.

Some concerns remained, which as I mentioned.

Capitalization of tables and figure titles is still inconsistent. I strongly recommend that you ask an academic writer to check the whole paper.

Major

1)Novelty needs to be discussed. It is not surprising that the cutoff changes when the population changes, and it has not changed significantly from previous reports.

The authors stated that it was a preliminary study, but on the other hand, they also said that this study was a validation study in point by point.

2) If looking at the number of events, it could be possible to adjust for summarized risk score. A scoring system has been proposed in several studies. (e.g. Abualhin, Mohammad et al. Journal of Vascular Surgery, Volume 70, Issue 3, 901 – 912.).

Minor

  1. The capitalization of the Figure title and title of sub-section is still inconsistent. The title of Figure 1 is "Enroll Flowchart", but Figure 2 does not capitalize the second and subsequent words.
  2. Abbreviations are used in Figure and Table, but they need to be self-explanatory.

Author Response

The manuscript has been much improved.

Some concerns remained, which as I mentioned.

We really thank the Reviewer for his/her comment, as we tried to solve any problem underlined by the Reviewer.

Capitalization of tables and figure titles is still inconsistent. I strongly recommend that you ask an academic writer to check the whole paper.

Capitalization of Table and Figures were deeply revised, as well as the entire document, to make capitalization more consistent, in line with Instruction for Authors.

Major

  • Novelty needs to be discussed. It is not surprising that the cutoff changes when the population changes, and it has not changed significantly from previous reports.

The authors stated that it was a preliminary study, but on the other hand, they also said that this study was a validation study in point by point.

We thank the Reviewer for his/her kind comments, and we fully agree with his/her consideration about cut-off change when population changes.

However, the aim and (we are strongly convinced) the novelty of present study was precisely to overcome this unavoidable equation: we tried to identify and preliminarily validate a unique cut-off that does not change when population changes, able to identify a subgroup of patients at higher risk for all the acute limb ischemia related outcomes (mortality, amputation, and a combination of them).

This is a crucial difference when comparing our results with the (only two) previously published papers on the same topic: our study validated and proposed a unique and simple cut-off (NLR >5), while other experiences just report NLR values found to be significative in their own study populations.

  • If looking at the number of events, it could be possible to adjust for summarized risk score. A scoring system has been proposed in several studies. (e.g. Abualhin, Mohammad et al. Journal of Vascular Surgery, Volume 70, Issue 3, 901 – 912.).

We thank the Reviewer for his/her kind comments. However, the article he/she refers to is about Critical Limb Ischemia (CLI) not about Acute Limb Ischemia (ALI).  Although both, CLI and ALI, are life- and limb-threatening conditions they are two completely different nosological entities that could not be assimilated, especially in management and clinical outcomes.

Consequently, we should admit that the score proposed in Bologna by Abualhin et all could not be used in case of Acute Limb Ischemia.

Minor

  • The capitalization of the Figure title and title of sub-section is still inconsistent. The title of Figure 1 is "Enroll Flowchart", but Figure 2 does not capitalize the second and subsequent words.

We think the Reviewer for his/her kind comments. Capitalization of Table and Figures were deeply revised, as well as the entire document, to make capitalization more consistent, in line with Instruction for Authors.

  • Abbreviations are used in Figure and Table, but they need to be self-explanatory.

We thank the Reviewer for his/her kind comments. Figures and tables’ abbreviations were clarified wherever needed.
